# Experimental and DFT Research on the Effects of O_2_/CO_2_ and O_2_/H_2_O Pretreatments on the Combustion Characteristics of Char

**DOI:** 10.3390/molecules28041638

**Published:** 2023-02-08

**Authors:** Lei Zhang, Jie Xu, Rui Sun, Zhuozhi Wang, Xingyi Wang, Mengfan Yuan, Jiangquan Wu

**Affiliations:** 1School of Energy Science and Engineering, Harbin Institute of Technology, Harbin 150001, China; 2School of Chemical Engineering and Technology, Hebei University of Technology, Tianjin 300401, China

**Keywords:** O_2_/CO_2_ pretreatments, O_2_/H_2_O pretreatments, char, DFT, combustion mechanisms

## Abstract

The use of a coal-based energy structure generates a large amount of CO_2_ and NOx. The numerous emissions from these agents result in acid rain, photochemical smog, and haze. This environmental problem is considered one of the greatest challenges facing humankind in this century. Preheating combustion technology is considered an essential method for lowering the emissions of CO_2_ and NO. In this research, the char prepared from O_2_/CO_2_ and O_2_/H_2_O atmospheres was employed to reveal the effects of the addition of an oxidizing agent on the combustion characteristics of char. The structural features and combustion characteristics of preheated chars were determined by Raman, temperature-programmed desorption (TPD), and non-isothermal, thermo-gravimetric (TGA) experiments. According to the experimental results, the addition of oxidizing agents promoted the generation of smaller aromatic ring structures and oxygen-containing functional groups. The improvement in the surface physicochemical properties enhanced the reactivity of char and lowered its combustion activation energy. Furthermore, the combustion mechanisms of the char prepared from the O_2_/CO_2_ and O_2_/H_2_O atmospheres were investigated using the density functional theory (DFT). The simulation results illustrated that the combustion essence of char could be attributed to the migration of active atoms, the fracture of the benzene ring structure, and the reorganization of new systems. The addition of oxidizing agents weakened the conjugated components of the aromatic ring systems, promoting the successive decomposition of CO and NO. The results of this study can provide a theoretical basis for regulating the reaction atmosphere in the preheating process and promoting the development of clean combustion for high-rank coals.

## 1. Introduction

Anthracite is widely distributed in China, accounting for approximately 11.5% of the explored coal reserves [1]. However, due to the low volatility and poor reactivity of anthracite, it inevitably suffers from the difficulties of ignition and burn-out [2], resulting in severe environmental problems such as photochemical smog, acid rain, forest deterioration, and ozone depletion [3]. Owing to the strict standards for NOx emissions in China (decreased below 50 mg/m^3^ at 6% O_2_) [4], numerous researchers have made great efforts to lower NOx emissions, including through selective catalytic reduction (SCR) [5], selective non-catalytic reduction (SNCR) [6], low-NOx burners [7], and air/fuel staged combustion technology [8]. However, these methods have inevitably exhibited limitations such as catalyst pollution, ammonia leaking, and high operational cost. Considering the economic factors and environmental benefits, preheating combustion technology has been gradually developed to improve the fuel quality of anthracite [9].

During the preheating combustion process, coal particles are initially preheated in a circulating, fluidized bed. The preheated coals and gaseous agents are then reacted in a down-fired combustor (DFC) [10,11]. After the preheating treatments, the combustion efficiency of pulverized coal is obviously enhanced, and the corresponding NOx emissions clearly decrease. Yao et al. [12] found that the NOx emissions in the combustion process of preheated char were approximately 50 mg/N m^3^ (at 6% O_2_), much lower than the traditional combustion mode (850–1300 mg/N m^3^ at 6% O_2_)) [13]. Ding et al. [14,15] optimized the preheating combustion technology by replacing O_2_/N_2_ with O_2_/CO_2_ and O_2_/H_2_O. The variations in the preheating atmospheres made the temperature profile in DFC more uniform. Owing to the differences in the physicochemical properties of H_2_O, CO_2_, and N_2_, the replacement of N_2_ with CO_2_ and H_2_O altered the properties of chars, enhancing the combustion efficiencies of the char [16,17]. Although these investigations decoupled the preheating and combustion processes of pulverized coal, the correlation between the physicochemical properties of char and its combustion characteristics was still unclear, and the corresponding combustion mechanisms still required further investigation.

Limited by the experimental means, the combustion characteristics of char still lack investigation. Therefore, the specific reaction mechanisms (such as active sites and intermediate and elementary reactions) of combustion reactions need to be studied through simulation. With the improvement of computers and the optimization of computing software, the reaction mechanisms, which are difficult to observe experimentally, can be determined using the DFT method. The emission mechanisms of CO and NO during the oxidization process of the nitrogen-containing char were determined by Zhang et al. [18] through the DFT method. They found that the first step for the oxidization of char was the adsorption of O_2_ on the surface of the char. Huo et al. [19] investigated the effects of H_2_O on the formation mechanisms of different oxygen-containing functional groups during spontaneous coal combustion. The calculated results demonstrated that the addition of the H_2_O molecule decreased the oxidation activation energy of ethylbenzene hydroperoxide. Owing to the significant effects of hydroxyl groups on the development of coal combustion, Zhu et al. [20] have investigated the reaction characteristics of hydroxyl groups during the coal combustion process. It was found that the hydrogen in a hydroxyl group was the active site of nucleophilic reaction. Additionally, the covalent bond C-H was more vulnerable to oxygen. Although the mechanisms of coal combustion have been studied by numerous scholars, the effects of oxidizing pretreatments on the reaction characteristics have not been considered.

Although many scholars have recently investigated the combustion characteristics of anthracite, these investigations were mainly conducted through the experimental method. Further exploration is needed for the detailed mechanisms of coal combustion. Moreover, considering the complexity of coal combustion with the participation of multiple reaction gases, little research was focused on the combustion mechanism of coal under the action of multiple gases. In this research, isothermal preheating experiments under various atmospheres (Ar, O_2_/CO_2_, and O_2_/H_2_O) were carried out in a horizontal tubular furnace. The physical and chemical properties of the surface, combustibility, and kinetic characteristics of the preheated chars were determined via Raman, TPD, and TGA experiments. The combustion mechanisms of char prepared from O_2_/CO_2_ and O_2_/H_2_O atmospheres were revealed by the DFT method. By coupling the macroscopic experimental and theoretical results, the effects of oxidizing preheating treatments on the char combustion were determined. The mutual confirmation of macroscopic experiment and microscopic mechanism provided a new method for coal science. Although there are some differences between this research and the actual high-rank coal preheating system, the results of this study may also provide a theoretical basis for regulating the reaction atmosphere in the preheating process, promoting the development of clean combustion for high-rank coals.

## 2. Experiments and Calculations

### 2.1. Preparation of Preheated Char

Jin Cheng (JC) anthracite was selected as the raw coal sample in this research. The corresponding proximate (HD-GF500, China Huadian Corporation, Beijing, China) and ultimate (Vario MACRO cube CHNS, Elementar, Langenselbold, Germany) analysis results are listed in Table 1. Prior to the preheating experiment, the anthracite was sieved into a diameter range of 90–120 μm and thoroughly dried for 48 h at 378 K. The preheating experiments were conducted in a horizontal tubular furnace system (as shown in Figure 1) under various atmospheres (Ar, O_2_/CO_2_, and O_2_/H_2_O). During each preheating experiment, the weight of anthracite was 500 mg. The detailed experimental conditions are illustrated in Table 2. The preheated chars, derived from various preheating processes, were collected for subsequent analyses.

### 2.2. Analytical Methods

#### 2.2.1. Non-Isothermal, Thermo-Gravimetric Experiments

Non-isothermal, thermo-gravimetric analysis has been widely used to test the reactivity of carbonaceous materials [21,22]. The combustion characteristics of preheated char (including ignition temperature, burn-out temperature, and comprehensive combustion index) were determined by TGA (Rigaku TG-DTA 8122). Approximately 6 mg of preheated char was placed in a pan for each TG test. The sample was then heated from room temperature to 378 K and held for 15 min to eliminate the effect of moisture content. The combustion atmosphere was 21 vol.% O_2_ balanced with 79 vol.% Ar. Subsequently, the preheated char was further heated to 1273 K at 10 K/min in the identical reaction atmosphere and maintained for 30 min until burn out.

The activation energy (*E*) and pre-exponential factor (*A*) are the two specific kinetic parameters during the combustion process of carbonaceous materials [23]. According to the Coats–Redfern integral method, the non-isothermal combustion characteristics of preheated chars can be well described by Equations (1) and (2):(1)$$\frac{d\alpha }{dt}=k(T)f(\alpha )$$
(2)$$k(T)=A\exp(-\frac{E}{RT})$$
where *α* is the mass conversion degree, *t* represents the reaction time (min), *k*(*T*) represents the rate constant, *A* is the pre-exponential factor (min^−1^), *E* is the activation energy (kJ/mol), R is the universal gas constant with the value of 8.314 J/(mol·K), and *T* is the absolute temperature (K).

Previous investigations have represented the combustion of coal-based fuel through a first-order reaction [24,25]. Therefore, the reaction order (n) is marked as 1, and *f* (*α*) can be presented as Equations (3) and (4):(3)$$f(\alpha )={\left(1-\alpha \right)}^{n}=1-\alpha$$
(4)$$\alpha =\frac{\omega_{0}-\omega_{t}}{\omega_{0}-\omega_{f}}$$
where ω_0_ and ω*_f_* are the initial and final weights of the pulverized char (mg), respectively, and ω*_t_* is the weight of char at the time of *t* (mg).

The heating rate (*β*) can be calculated through Equation (5):(5)$$\beta =\frac{dT}{dt}$$

Therefore, Equation (1) can be rewritten as Equation (6) by combining the Equations (2) and (5):(6)$$\frac{d\alpha }{dT}=\frac{1}{\beta }\frac{d\alpha }{dt}=\frac{1}{\beta }A\exp(-\frac{E}{RT})(1-\alpha )$$

Equation (7) can be obtained by integrating Equation (6), taking logarithms and simplifying treatments [26]:(7)$$\ln[\frac{-\ln(1-\alpha )}{T^{2}}]=\ln(\frac{AR}{\beta E})-\frac{E}{RT}$$

It is evident that the plot of $\ln(\frac{-\ln(1-\alpha )}{T^{2}})$ versus 1/*T* is a straight line. Therefore, *E* and *A* can be determined by the slope and the intercept of the regression line.

#### 2.2.2. Raman Analysis

The first-order Raman spectrum from 800 to 1800 cm^−1^ was analyzed in this research. The spectrum in this area was fitted to ten characteristic bands according to the methodology proposed by Li et al. [27]. Detailed information on the band can be found in previous research [28]. Generally, the D band in the Raman spectrum corresponds to the medium-to-large structure in preheated char (≥6). The G_r_, V_L_, and V_r_ bands are the typical amorphous carbon structures with 3–5 benzene rings. The ratio I_(Gr + VL + Vr)_/I_D_ is usually used to reflect the relative amount of small aromatic rings in carbonaceous materials.

#### 2.2.3. Temperature-Programmed Desorption Experiments

In order to determine the amounts of C(O) on the surface of char, TPD experiments were carried out in the horizontal tubular furnace. A corundum boat loaded with approximately 50 mg of dried char was placed in the constant-temperature section of the furnace. The sample was then heated to 1723 K under a pure Ar atmosphere with a heating rate of 5 K/min. The volume flow rate of the reaction gas was 1.0 L/min. By employing an IR gas analyzer (Gasboard-3000, China), the volumetric fractions of CO and CO_2_ in the exhaust were recorded. The amounts of CO and CO_2_ released from each char can be calculated using Equations (8) and (9).
(8)$$n_{CO}=\frac{1000\times Q_{TPD}\times \int_{0}^{t}(C_{CO}\times 10^{-6})dt}{22.4\times 60\times m}$$
(9)$$n_{CO}{}_{{}_{2}}=\frac{1000\times Q_{TPD}\times \int_{0}^{t}(C_{CO_{2}}\times 10^{-6})dt}{22.4\times 60\times m}$$
where *n*_CO_ and
$n_{CO}{}_{{}_{2}}$ are the total amounts of CO and CO_2_ generated from each pretreated char (μmol/g), respectively; *Q*_TPD_ reflects the volume flow rate of the reaction gas (L/min); *t* is the reaction time of each TPD test (s); *C*_CO_ and $C_{CO_{2}}$ represent the volumetric fractions of CO and CO_2_ (ppm) during each TPD experiment (ppm), respectively; and *m* is the mass of each char (mg).

### 2.3. Calculated Details

In this research, the DFT method was employed to reveal the combustion mechanisms of char derived from oxidizing torrefaction. It is widely known that the reactivity of char is relevant to the type of reacting edge, and that the zigzag edges are more active than armchair edges [29,30,31]. Therefore, a single layer of graphite consisting of seven rings with zigzag edges was selected to simulate the initial model of preheated char (as shown in Figure 2a). The initial model was optimized considering the accumulated phenomenon of nitrogen elements [32]. The corresponding configuration is shown in Figure 2b. The optimized structures, transition states, and zero point energy were obtained at the level of M062X/6-31G(d,p). The transition states were all verified by the intrinsic reaction coordinate calculation (IRC). All calculations were performed with Gaussian 09 [33], and the wave function was analyzed using Multiwfn [34].

## 3. Results and Discussion

### 3.1. Experimental Results and Discussions

#### 3.1.1. Characterization of the Physical and Chemical Properties of Preheated Char

It is widely known that compositional variation and elemental distribution plays a vital role in determining the quality of fuel [35]. The amounts of fixed carbon expressed a positive effect on the energy density of preheated char, and a higher fixed carbon amount corresponded to a higher fuel quality [36]. Therefore, the physical and chemical properties of each char are summarized in Figure 3. According to the data shown in Figure 3a, the addition of oxidizing agents enhanced the consumption of volatile content, leading to the accumulation of fixed carbon content in char particles. The variant characteristics of fixed carbon were relevant to the preheating conditions. For the O_2_/CO_2_ preheating treatments, the fixed carbon content of the preheated sample reached its maximum value after undergoing torrefaction performed in an atmosphere of 6 vol.% O_2_ + 30 vol.% CO_2_, illustrating that the enhancement of CO_2_ under identical O_2_ volumetric fraction was beneficial to the accumulation of fixed carbon content. For the O_2_/H_2_O preheating treatments, the further enhancement of H_2_O was beneficial to the consumption of fixed carbon. Affected by the gasification between H_2_O and carbon, the excessive H_2_O molecules wrapped around the char particles, inhibiting further oxidization and gasification among the O_2_, H_2_O, and coal particles and leading to a decrease in the quality of the char [22].

Raman spectroscopy was widely employed to investigate the relationship between the microstructural features and the preheating conditions [27,37]. Generally, I_(Gr+VL+Vr)_/I_D_ and C(O) were the two vital indexes for characterizing the reactivity of carbonaceous materials. Therefore, the values of the I_(Gr+VL+Vr)_/I_D_ and C(O) amounts as two functions of CO_2_ and H_2_O volumetric fractions are exhibited in Figure 3b. The experimental results illustrated that the I_(Gr+VL+Vr)_/I_D_ and C(O) amounts of chars derived from O_2_/CO_2_ pretreatments and O_2_/H_2_O pretreatments were significantly higher than those preheated in the inert atmosphere. For the O_2_/CO_2_ pretreatments, the enhancement of the I_(Gr+VL+Vr)_/I_D_ and C(O) amounts can be attributed to the reactions between the oxidizing agents and the cross-linking systems in anthracite particles [38]. These reactions inhibited the combination of aromatic ring clusters and promoted the generation of new active sites. In the process of O_2_/H_2_O pretreatment, the amounts of C(O) and I_(Gr+VL+Vr)_/I_D_ for each char were relevant to the amount of additional H_2_O. When the volumetric fraction of H_2_O was relatively low, the attachment of oxygen free radicals (·O) to active sites inhibited the condensation of aromatic systems and promoted the generation of new reactive structures [37], exerting positive effects on the enhancement of C(O) and I_(Gr+VL+Vr)_/I_D_. When the volumetric fractions of H_2_O further increased, the decomposition of H_2_O molecules promoted the generation of ·H [11]. These radicals permeated the interior matrix of the char and promoted the condensation of small ring structures [22], leading to the decrease in I_(Gr+VL+Vr)_/I_D_ [11]. Although high volumetric fractions of H_2_O accelerated the consumption of partial, smaller aromatic rings, the total amount of reactive structures was still relatively high. The corresponding I_(Gr+VL+Vr)_/I_D_ values were 1.70 for 10 vol.% of H_2_O, 1.64 for 20 vol.% of H_2_O, and 1.49 for 30 vol.% of H_2_O.

#### 3.1.2. Combustion Characteristics of Pretreated Char

The TG−DTG method was employed to identify the combustion characteristics of chars derived from various pretreatment conditions. As is shown in Figure 4a, the combustion characteristic parameters of the TG and DTG curves included the ignition temperature (*T*_i_), the burn-out temperature (*T*_b_), the maximum combustion rate ((*dω*/*dt*)_max_), and the average mass loss rate ((*dω*/*dt*)_mean_). Therefore, the comprehensive combustion index (*S*) for each sample can be determined through Equation (10). *S* was generally employed to evaluate the combustion characteristics of carbonaceous materials, which expressed a positive correlation with the combustibility of char [26].
(10)$$S=\frac{{\left(dw/dt\right)}_{\max}{\left(dw/dt\right)}_{mean}}{T_{i}^{2}T_{b}}$$

According to the experimental results, the combustion parameters of each char are summarized in Figure 4b and Table A1. Compared with the inert pretreatment, the addition of the oxidizing agents showed little effect on the ignition and burn-out. The preheated chars derived from O_2_/CO_2_ atmospheres slightly decreased the ignition and burn-out temperatures. The synergy between the O_2_ and CO_2_ agents accelerated the accumulation of volatile matters around char particles, expressing little positive effect on the ignition and burn-out of the preheated chars, while the effects of H_2_O on the ignition temperatures of chars demonstrated an opposite tendency. *S* was generally considered the essential parameter for reflecting the combustion characteristics of char [39]. According to the experimental results shown in Figure 4b, the value of *S* for the char preheated in the Ar atmosphere was 6.69, much lower than the other pretreatments, illustrating that the O_2_/CO_2_ and O_2_/H_2_O pretreatments enhanced the combustibility of chars. The positive effects of the O_2_/CO_2_ pretreatments on the combustibility of the chars might be the oxidization and gasification reactions of the chars [22]. For the O_2_/H_2_O preheating processes, numerous ·OH and H radicals were generated and remained in the char particles. These radicals were all had a high penetrating ability, and could infiltrate the matrix of char to promote the formation of a fragment structure [23]. These stabilized species were difficult to ignite but easy to burn out [40]. Although the O_2_/H_2_O pretreatments delayed the ignition of chars, O_2_/H_2_O pretreatments were more conducive to the combustion of chars than Ar and O_2_/CO_2_ pretreatments.

The activation energy and pre-exponential factor were calculated to evaluate the effects of O_2_/CO_2_ and O_2_/H_2_O pretreatments on the combustion reactivity of chars. Zhu et al. [41] demonstrated that the combustion of pulverized coal included oxygen diffusion, oxygen adsorption, structural rearrangement, and functional group desorption. Song et al. [25] found that the kinetic data derived from the low-temperature region were more conducive to characterizing the intrinsic kinetics. Therefore, Arrhenius plots of the chars in the temperature range of 743 K–833 K were calculated, and the results are summarized in Figure A1. According to the data shown in Table 3, it could be determined that the activation energy of the char preheated in the Ar atmosphere was 46.95 kJ/mol, much higher than the activation energies of the chars preheated in the O_2_/CO_2_ and O_2_/H_2_O atmospheres. This phenomenon can be attributed to the improvement in anthracite quality through oxidizing pretreatments [22]. The activation of oxidizing agents on anthracite particles improved the structural characteristics and the surface chemistry of chars. These variations decreased the combustion activation energy of char. At the same oxygen fraction, the positive effects of CO_2_ on char combustion increased with the volumetric fraction of CO_2_. However, the positive effect of H_2_O on the combustibility of char decreased with the volumetric fractions of H_2_O. The appropriate preheating condition for anthracite was 6 vol.% O_2_ + 10 vol.% H_2_O. This result was strongly consistent with the comprehensive combustion index and Raman results.

### 3.2. Theoretical Investigation of the Combustion Mechanisms of Preheated Char

#### 3.2.1. Combustion Characteristics of Pretreated Char

Previous investigation demonstrated that the addition of oxidizing agents during the preheating treatments promoted the generation of oxygen-containing functional groups [32]. These groups included phenol, quinone, carboxyl, lactone, and anhydride. Among them, the amount of phenol groups was the highest [11]. Therefore, the charN model decorated with one phenol group was used to simulate the structure of char prepared from O_2_/CO_2_ and O_2_/H_2_O preheating treatments (which can be abbreviated as charNOH). The corresponding configuration is shown in Figure 5a. The atomic dipole corrected Hirshfeld atomic charge (ADCH) analysis was employed to describe the structural characteristics of char. As shown in Figure 5b,c, the atoms N_2_, C_4_, and C_8_ in charN were all negatively charged, and the atom C_6_ was positively charged. When the char was decorated with the phenol group, the electronic characteristics of the atoms C_6_ and C_8_ were altered. The electron of atom C_6_ changed from positive to negative, and atom C_8_ changed from negative to positive. This meant that the additional phenol group altered the physical and chemical properties of char, which could alter the subsequent combustion process through affecting the reactivity of char.

#### 3.2.2. The Combustion Mechanisms of Pretreated Char

There are no excellent observational means that can capture the component transformation from an atomic-level perspective. Therefore, the combustion mechanisms of preheated chars were still unclear. In this research, the DFT method was used to attempt to reveal the combustion mechanisms of char prepared from O_2_/CO_2_ and O_2_/H_2_O pretreatments. The optimized structures of the stable states and transition states for the combustion of char derived from O_2_/CO_2_ pretreatments are shown in Figure A2 and Figure A3, and the corresponding energy potential diagram is shown in Figure 6. For the O_2_/CO_2_ preheating treatments, the initial step in the combustion of the char was the adsorption of CO_2_. Our previous investigation illustrated that the adsorption sites of CO_2_ were the atoms C_4_ and C_6_ [42]. Thence, when the CO_2_ molecule approached the surface of char, the adsorption of CO_2_ occurred immediately. This process was a low-energy barrier step: the corresponding barrier value was 10.27 kJ/mol. Once the adsorption of CO_2_ finished, the decomposition of CO occurred. The formation of IM2 was a relatively low-energy barrier step. The barrier value was 235.08 kJ/mol. IM2 was a crucial structure, connecting the two combustion pathways of the char (Path 1 and Path 2). When the combustion of the char occurred in the form of IM2, the reaction process might obey the route of Path 1. In this pathway, the adsorption of O_2_ occurred on the sites of N_2_ and C_4_ atoms. Subsequently, the successive decomposition of CO and NO occurred through the process of $\mathrm{IM}3\overset{\mathrm{TS}4}{\to }\mathrm{IM}4\overset{\mathrm{TS}5}{\to }\mathrm{IM}5\overset{\mathrm{TS}6}{\to }\mathrm{IM}6\overset{\mathrm{TS}7}{\to }\mathrm{IM}7$. After the formation of IM7, the migration of the H_1_ atom from O_1_ to C_9_ occurred. This process was a low-energy barrier step: the corresponding value was 134.69 kJ/mol. Then, the decomposition of the last CO molecule happened through the process of $\mathrm{IM}8\overset{\mathrm{TS}9}{\to }\mathrm{IM}9\overset{\mathrm{TS}10}{\to }\mathrm{IM}10$. The final step of ***Path 1*** was the reorganization of aromatic rings ($\mathrm{IM}10\overset{\mathrm{TS}11}{\to }P1$). This process was the rate-determining step of Path 1. The corresponding energy gap was 1179.39 kJ/mol.

Different from Path 1, prior to the combustion, the structure of char was reorganized in Path 2. Firstly, the H_1_ atom migrated from O_1_ to C_9_, and the successive decomposition of CO molecules then occurred through the process of $\mathrm{IM}3'\overset{\mathrm{TS}4'}{\to }\mathrm{IM}4'\overset{\mathrm{TS}5'}{\to }\mathrm{IM}5'\overset{\mathrm{TS}6'}{\to }\mathrm{IM}6'\overset{\mathrm{TS}7'}{\to }\mathrm{IM}7'$. Subsequently, the cyclization of the aromatic ring occurred. This process was a low-energy barrier step: the barrier value was 25.84 kJ/mol. The combustion of the remaining char then occurred. According to the different decompositions of NO and CO, the remaining route of Path 2 consisted of two pathways. If the decomposition of CO occurred prior to the NO molecule, the remaining pathway might obey Path 2, and the process of $\mathrm{IM}9'\overset{\mathrm{TS}10''}{\to }P2$ was the rate-determining step of Path 2. The energy gap was 1275.19 kJ/mol. When the decomposition of NO occurred before the decomposition of CO, the subsequent reaction might obey the branch of Path 2. Owing to the lower barrier value of $\mathrm{IM}9'\overset{\mathrm{TS}10'}{\to }\mathrm{IM}10'$ when compared to $\mathrm{IM}9'\overset{\mathrm{TS}10''}{\to }P2$, Path 2 was more likely to occur than the branch of Path 2. Considering the lower energy gap in Path 1 than Path 2, Path 1 was considered the most possible route for the combustion of char prepared from O_2_/CO_2_ pretreatments.

For the O_2_/H_2_O preheating treatments, the optimized structures of stable states and transition states are summarized in Figure A4 and Figure A5. The corresponding energy potential diagram is shown in Figure 7. When the H_2_O molecule approached the surface of the char, IM1 was formed immediately. The energy barrier of this process was much lower, and its value was 15.10 kJ/mol. The migration of H_5_ and H_1_ atoms then successively occurred through the processes of $\mathrm{IM}1\overset{\mathrm{TS}2}{\to }\mathrm{IM}2\overset{\mathrm{TS}3}{\to }\mathrm{IM}3\overset{\mathrm{TS}4}{\to }\mathrm{IM}4$. After the formation of IM4, the successive decomposition of CO and NO occurred through the processes of $\mathrm{IM}4\overset{\mathrm{TS}5}{\to }\mathrm{IM}5\overset{\mathrm{TS}6}{\to }\mathrm{IM}6\overset{\mathrm{TS}7}{\to }\mathrm{IM}7\overset{\mathrm{TS}8}{\to }\mathrm{IM}8$. The allotropy of IM8 was then formed through the process of $\mathrm{IM}8\overset{\mathrm{TS}9}{\to }\mathrm{IM}9$. This process was a low-energy barrier step: the corresponding value was 65.19 kJ/mol. After the migration of the H_4_ atom, the successive decomposition of CO molecules occurred. The process of $\mathrm{IM}11\overset{\mathrm{TS}12}{\to }\mathrm{IM}12$ was considered the rate-determining step of ***Path 3***, and the corresponding energy gap was 1119.35 kJ/mol. Compared with the O_2_/CO_2_ pretreatments, the replacement of CO_2_ with H_2_O optimized the pathway of char combustion. It was determined that the essence of char combustion was the migration of active atoms, the fracture of the benzene ring structure, and the reorganization of new systems.

## 4. Conclusions

In this research, the combustion mechanisms of char prepared from O_2_/CO_2_ and O_2_/H_2_O pretreatments were revealed by the employment of experimental and DFT methods. The experimental results demonstrated that the addition of oxidizing agents enhanced the consumption of volatile content, leading to the accumulation of fixed carbon content in char particles. Moreover, the addition of oxidizing agents promoted the generation of oxygen-containing functional groups and small aromatic ring structures. The improvement of surface physicochemical properties enhanced the reactivity of char and lowered its combustion activation energy. Based on the experimental results shown in this research, the conditions of 6 vol.% O_2_ + 30 vol.% CO_2_ and 6 vol.% O_2_ + 10 vol.% H_2_O were the most appropriate conditions for preparing the char particles. The DFT method was also employed to further reveal the mechanisms of char combustion. The theoretical results demonstrated that the combustion essence of char was the migration of active atoms, the fracture of benzene ring structure, and the reorganization of new systems. The addition of oxidizing agents altered the combustion process of char by affecting the physical and chemical properties of the char. Compared with the O_2_/CO_2_ pretreatments, the O_2_/H_2_O preheating treatments made the char more active, and the replacement of CO_2_ with H_2_O optimized the combustion pathway of the char.

## Figures and Tables

**Figure 1 molecules-28-01638-f001:**
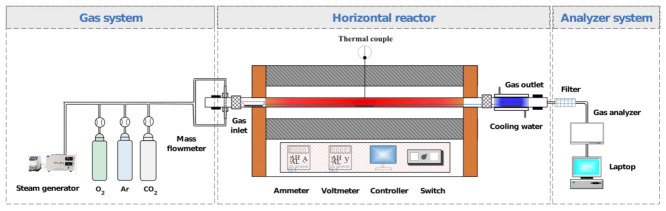
Schematic of the horizontal tubular furnace.

**Figure 2 molecules-28-01638-f002:**
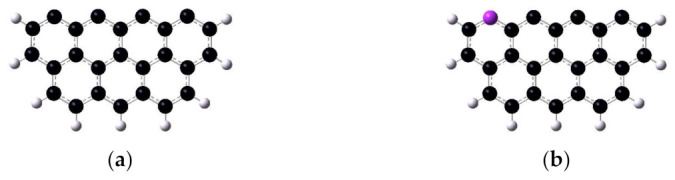
The optimized structure of typical models. (**a**) The carbonaceous surface, abbreviated as char. (**b**) The carbonaceous surface decorated with one pyridine group, abbreviated as charN.

**Figure 3 molecules-28-01638-f003:**
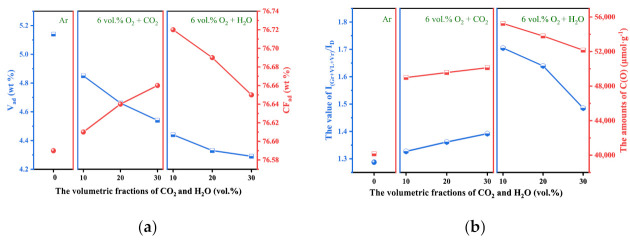
Physical and chemical properties of preheated char. (**a**) The volatile content and fixed carbon content remained in pulverized char. (**b**) The ratios I_(Gr+VL+Vr)_/I_D_ and C(O) amounts on the surface of char.

**Figure 4 molecules-28-01638-f004:**
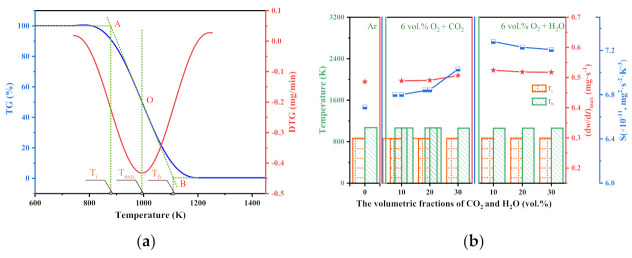
TG−DTG results. (**a**) The TG and DTG curves of Example 1. (**b**) The characteristic combustion parameters of each char.

**Figure 5 molecules-28-01638-f005:**
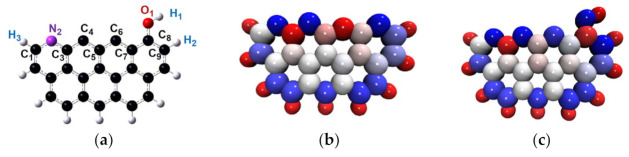
The configuration of preheated char. (**a**) The initial structure of char prepared from O_2_/CO_2_ and O_2_/H_2_O pretreatments, abbreviated as charNOH. (**b**) The ADCH of charN. (**c**) The ADCH of charNOH.

**Figure 6 molecules-28-01638-f006:**
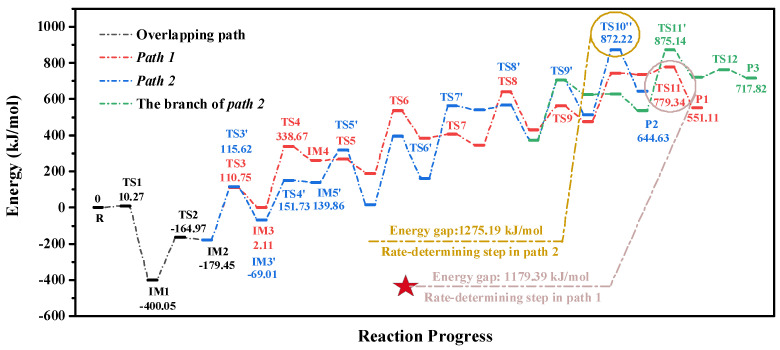
Energy potential diagram for the combustion of char prepared from O_2_/CO_2_ atmospheres.

**Figure 7 molecules-28-01638-f007:**
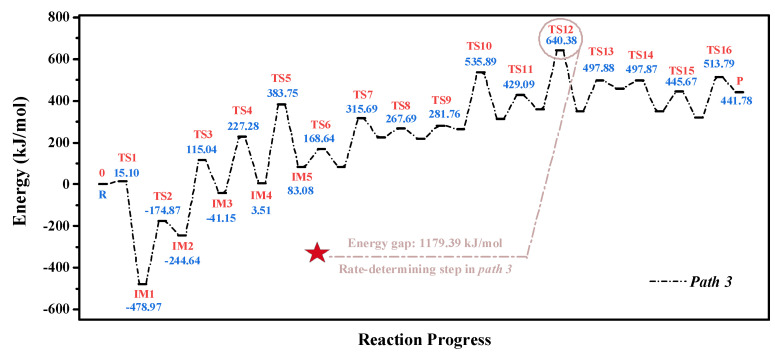
Energy potential diagram for the combustion of char prepared from O_2_/H_2_O atmospheres.

**Table 1 molecules-28-01638-t001:** Ultimate and proximate analysis of raw coal sample.

Samples	Ultimate Analysis (Dry and Ash-Free)					Proximate Analysis (as Received)			
	C (wt%)	H (wt%)	N (wt%)	S (wt%)	O (wt%)	Moisture (wt%)	Volatiles (wt%)	C_fixed_ * (wt%)	Ash (wt%)
JC	90.23	3.22	1.05	1.35	4.15	2.51	7.84	76.54	13.11

* Determined by the difference.

**Table 2 molecules-28-01638-t002:** Preheating experimental conditions.

Parameter	Temperature	Residence Time	Atmosphere
Sample 1	1173 K	25 s	Ar
Sample 2	1173 K	25 s	6 vol.% O_2_ + 10 vol.% CO_2_ + Ar balanced
Sample 3	1173 K	25 s	6 vol.% O_2_ + 20 vol.% CO_2_ + Ar balanced
Sample 4	1173 K	25 s	6 vol.% O_2_ + 30 vol.% CO_2_ + Ar balanced
Sample 5	1173 K	25 s	6 vol.% O_2_ + 10 vol.% H_2_O + Ar balanced
Sample 6	1173 K	25 s	6 vol.% O_2_ + 20 vol.% H_2_O + Ar balanced
Sample 7	1173 K	25 s	6 vol.% O_2_ + 30 vol.% H_2_O + Ar balanced

**Table 3 molecules-28-01638-t003:** Kinetic parameters calculated by the Coats–Redfern method.

Samples	1	2	3	4	5	6	7
*E* (kJ/mol)	46.95	39.08	38.78	30.62	27.15	27.29	27.48
*A* × 10^−7^ (min^−1^)	1.62	4.37	4.53	10.20	16.10	15.60	14.20
R^2^	0.99	0.99	0.99	0.99	0.98	0.99	0.99

## Data Availability

The authors do not have permission to share data.

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
