# Peer review of "Experimental and DFT Research on the Effects of O2/CO2 and O2/H2O Pretreatments on the Combustion Characteristics of Char"

_molecules, 2023, doi:10.3390/molecules28041638_

Round 1
Reviewer 1 Report
Comments to the authors
1. It is advisable to avoid the use of abbreviations, even relatively well-known, without explaining them at the point of the first mention (DFT, TPD, TGA).
2. The work deals with the effects of the addition of oxidising agents on the combustion characteristics of semi-char. Taking into consideration its focus and content, thematically, the proposed manuscript can be regarded as relevant to the wider scope of the journal.
3. The title correctly describes the manuscript's content. The keywords are adequately selected.
4. To properly represent the essence of the conducted research, the abstract should briefly cover the following:
- context, background and motivation for undertaking the research,
- hypothesis,
- the method(s) used,
- the main results obtained,
- the conclusion(s) drawn, emphasising the implications of the main findings.
In the present abstract version, most of the required components are present, but some are not. This means that the authors should slightly reorganise and improve the text of the abstract.
5. The overall structure of the manuscript is created in accordance with the recommendations for a typical scientific paper.
6. The literature review can be considered comprehensive and properly done, but the authors use lumped references on several occasions, which should be avoided.
7. In the last paragraphs of Section 1 (Introduction), the authors explain what the core of their research is. However, they should clearly state the gap they try to fill in the respected scientific area and their contribution that makes the research different from the works of other authors. They need to elaborate further on the extent of scientific novelty in their research.
8. The methodology of the research has been presented in Section 2, and it has been done quite correctly.
9. Taking into account the relatively wide range and complexity of the analysed issues, the presented results can be considered relevant. The discussion and elaboration of the results are adequate and quite comprehensive.
10. Despite the fact that the authors are mentioning what is the main contribution of their work (in the final paragraphs of Section 1!), they should further emphasise the extent of novelty and added value.
11. The conclusions are adequate and supported by the obtained and presented results.
12. The quality of some figures should be improved.
13. The style and wording of the manuscript are relatively correct.
Reviewer 2 Report
The authors of the manuscript entitled "Experimental and DFT research on the effects of O2/CO2 and O2/H2O pretreatments on the combustion characteristics of semichar" study, both experimentally and in DFT simulations, the effects of coal pre-heating in oxidative and inert atmospheres on some physicochemical parameters involved in coal combustion. The subject of the manuscript is interesting from the viewpoint of maximizing the coal combustion efficiency.
However, there are a number of issues that need to be addressed before the manuscript could be recommended for publication.
1. The term "semichar" is not generally accepted and not readily available from the literature. The authors should be aware that "char" means "charcoal". At the same time, it unambiguously follows from the manuscript that the object under study is anthracite, i.e. mined coal. This casts serious doubts on the appropriateness/relevance of the term "semichar" introduced by the authors as it may be misleading. The authors should consider choosing another, more appropriate term to define this product.
2. In Abstract, unexplained abbreviations should be avoided.
3. L.33-34: The units mg/N m3 should be replaced with mg N/m3 or mg N m-3
4. Which amounts (weights) of anthracite could be used in the preheating experiments?
5. L.98: What does "conversion" mean in this context? Does it mean combustion?
6. What is n in Eq. (3)? Why is it equated to 1? Why does n again appear in Eq. (6) and disappear in the integration?
7. L.110: From (7), it does not follow that the plot of ln (AR/βE) versus 1/T is a straight line. This could have been true if the left-hand side had had no temperature variable. For the sake of clarity, at least one explained example of such a plot should be provided in Results.
8. L.115-119: When characterizing the Raman spectra and their specific bands, the authors cite the work [35]. However, this reference does not contain any information relevant to this point. Either the spectral characteristics of the D, Gr, VL and Vr bands or a relevant reference should be provided in the revision.
9. L.146: The section title needs to be changed to Results and Discussion.
10. It is not clear which parameters are given in the Y axes in Figure 3a. The respective information should be added to the caption.
10. Contrary to the text accompanying Figure 4 in the manuscript, there are no discernible differences between the ignition and burn-out temperatures of samples preheated in the inert and oxidizing atmospheres in this Figure.
11. How can the authors explain the fact that the energies of transition between the initial and final DFT-simulated states are at least one order of magnitude higher than the experimentally found activation energies of combustion?
12. In addition, there are some language issues/typos highlighted below that should be corrected.
L.41,308: semichar
L.84: spontaneous coal combustion
L.85-86: mixed use of grammar tenses (Past Simple and Present Perfect; the former is more suitable)
L.90: mechanisms of coal combustion
L.148 and thereinafter: What is meant by "physicochemical performance"? Either this not generally known term should be defined at the first mention or another more appropriate wording should be applied throughout the manuscript.
L.212: difficult to ignite
L.244-245: negatively charged, positively charged
L.252: that can capture
L.261: approached the surface
L.303,324: the pathway of semichar combustion
L.304,319: the essence of semichar combustion
L.318: the DFT method was employed to further reveal the mechanisms of semichar combustion
Figures 6, 7: "Reaction Progress" in the X Axes
I suggest that the authors should choose more appropriate words instead of "exerted" (L.203), "under" (L.229), "conducted" (L.240).
Round 2
Reviewer 2 Report
The authors have replied to all my questions and comments, and their response seems satisfatory to me. I would only suggest the following corrections.
L.79: were mainly conducted
L.80: further explorations were still needed
L.82: little research was focused on
L.85: physical and chemical properties of the surface
L. 114: identify->test
L. 238: addition of the oxidizing agents showed little effects on the ignition and burn-out
